# Synergistic Effect of Sic Particles and Whiskers on the Microstructures and Mechanical Properties of Ti(C,N)-Based Cermets

**DOI:** 10.3390/ma15062080

**Published:** 2022-03-11

**Authors:** Dunlei Yan, Guangtao Xu, Zhenhua Yao, Huachen Liu, Yikun Chen

**Affiliations:** 1School of Material Science and Engineering, Wuhan University of Technology, Wuhan 430070, China; ydl@whut.edu.cn (D.Y.); xgt9854@163.com (G.X.); 2New Tobacco Products Engineering Center, China Tobacco Hubei Industrial Co., Ltd., Wuhan 430070, China; v_jra2j2@163.com

**Keywords:** Ti(C,N)-based cermet, SiC particles, SiC whiskers, synergistic strengthening and toughening

## Abstract

The microstructure and mechanical properties of Ti(C,N)-based cermets with the addition of the SiC particles (SiC_p_) and SiC whiskers (SiC_w_), were systematically studied in this work. Firstly, the effect of SiC_p_ on the cermets was investigated independently to determine the considerable total amounts of additives, and the results showed that 2.0 wt.% SiC_p_ would lead to optimal properties of the cermet. Then, the influence of SiC_p_ and SiC_w_ additions with the variable ratio on the cermets was studied. The results indicated that when 1.5 wt.% SiC_p_ and 0.5 wt.% SiC_w_ were added; the cermets appeared with the best comprehensive properties, and the transverse rupture strength, hardness, and the fracture toughness of the cermets reached 2520.8 MPa, 88.0 HRA, and 16.56 MPa·m^1/2^, respectively. This was due to the synergistic strengthening and toughening effect afforded by the reasonable SiC_p_ and SiC_w_ addition, from which the smallest grain size, as well as the most uniform, and completed core-rim structure of the cermets, were achieved.

## 1. Introduction

Ti(C,N)-based cermets were widely applied in mechanical processing, metallurgy, and oil drilling fields as cutting tools and wearing parts, due to the excellent red hardness, wear resistance, and considerable strength and toughness. It was also identified as the substitution of cemented carbide, as the higher hardness and wear resistance allowed for more stable chemical properties, a stronger corrosion resistance, oxidation resistance, and smaller density [1,2].

However, the transverse rupture strength and fracture toughness of Ti(C,N)-based cermet are low, which limits its further application. Many scholars have adopted a variety of methods to improve the transverse rupture strength and fracture toughness of Ti(C,N)-based cermets, such as particle toughening, whisker toughening, etc. Particle toughening is the use of fine grain strengthening and crack deflection to achieve the effect of strengthening and toughening. Zhang et al. [3] investigated the effect of a nano-Si_3_N_4_ addition on the mechanical properties of Ti(C,N)-based cermet prepared by microwave sintering, and reported that the hardness and transverse rupture strength of cermet was improved by about 1.3% and 25.2% with a 2.0 wt.% Si_3_N_4_ addition. SiC_w_ has good chemical stability, a low coefficient of thermal expansion, and excellent wear resistance, and can be used as high-temperature-resistant materials and engineering materials, especially in the ceramic composite material reinforcement phase. Wu et al. [4] reported that when 1.0 wt.% SiC_w_ was added into the cermets, the highest transverse rupture strength and fracture toughness were increased by about 24% and 29%, respectively. The toughening mechanisms were characterized by crack deflection, whisker bridging, and whisker pulling-out.

The adding of second phase particles can be uniformly dispersed in the ceramic matrix, effectively refining the grains, and have an obvious strengthening effect on the matrix; however, the toughening effect is limited. Although the addition of whiskers can improve the toughness of the ceramic matrix, its strengthening effect on the matrix is not very obvious. The main reason is that the whiskers are not easily uniformly dispersed in the ceramic matrix. When the whisker amounts added are small, the effect of strengthening and toughening cannot be achieved. However, the excessive addition of whiskers will result in a decrease in density, which is not conducive to the mechanical properties of the ceramic matrix. Given the characteristics of particle and whisker toughening, both are used to strengthen and toughen synergistically; the effective and feasible way to improve the properties of ceramic was via synergistic strengthening and toughening, which has been widely applied in Al_2_O_3_ and ZrB_2_ matrix materials. Zhao et al. [5] revealed that TiC nanoparticles and SiC_w_ addition in hot-pressed Al_2_O_3_ ceramic would cause transgranular fracture and crack deflection, which improved the transverse rupture strength and fracture toughness. Pazhouhanfar et al. [6] added SiC_p_ and SiC_w_ into the ZrB_2_ matrix and found that the enhancing effect was more significant when particles and whiskers both existed and were synergistically performed.

However, there are few research reports on the application of the synergistic strengthening–toughening method to Ti(C,N)-based cermet. In this paper, the effect of SiC_p_ addition on the properties of cermet was firstly investigated systematically to decide the most considerable amounts of additives. Then, SiC_w_ and SiC_p_ with different ratios were added into the cermets to research the synergetic mechanism role.

## 2. Experimental Procedures

### 2.1. Sample Preparation

A mixture of TiC (2.58 µm), TiN (14.89 µm), Ni (2.18 µm), Mo (2.80 µm), WC (4.68 µm), Cr_3_C_2_ (3.79 µm), graphite (3.50 µm), SiC_p_ (45 µm) and SiC_w_ (length 10–40 µm, diameter 0.1–0.5 µm) was prepared by ball milling, as designed by (34–X)TiC-M-X wt.% SiC_p_ (X = 0, 1, 2, 2.5, 3), which was denoted as cermet A–E, and 32TiC-M-Y wt.% SiC_p_-(2–Y) wt.% SiC_w_ (Y = 0, 0.5, 1, 1.5), which were denoted as cermet F~I. The composition of M is listed in Table 1, and the morphologies of SiC_p_ and SiC_w_ are shown in Figure 1. The milling duration, milling rotation, and ball-to-powder were 48 h, 250 rpm, and 7:1, respectively, and the milling medium was ethanol. SiC_w_ was firstly dispersed by ultrasonication in alcohol and polyethylene for 1 h and then mixed with the milled powders for a further 2 h. After drying and sieving the mixture, the powders were compressed into green compacts with dimensions of 24 mm × 7 mm × 20 mm under the pressure of 200 MPa for 1 min. The green compacts were then sintered at 1450 ℃ in vacuum for 1 h.

### 2.2. Material Characterization

The microstructure of Ti(C,N)-based cermet was observed with the scanning electron microscope (SEM, JSM-IT300, JEOL Inc., Tokyo, Japan), and the grain size of Ti(C,N)-based cermet was tested from the SEM images by ImageJ software (ImageJ1.0, National Institutes of Health, Bethesda, MD, USA). The phase compositions were characterized by an X-ray diffractometer (XRD, D8 Advance, Bruker, Karlsruhe, Germany). The relative density of Ti(C,N)-based cermet was determined by the Archimedes method. The transverse rupture strength (TRS) of cermet was tested by a three-point bending method on an electronic universal material testing machine (Instron 5967, Laizhou Keyi Test Instrument, Laizhou, China) with a 1 mm/min loading speed. The hardness of the cermet specimen was measured through a standard Rockwell hardness tester (Micro-586, Shanghai Qingji Instrument Technology, Shanghai, China) at a constant load of 60 kg and a 10 s dwell time. The fracture toughness (K_IC_) of the sintered cermets was tested by using a single-edge notched bend (SENB).

## 3. Results and discussion

### 3.1. Effect of Sic_p_ on Cermets

The XRD patterns of cermets prepared with different amounts of SiC_p_ are shown in Figure 2a. Diffraction peaks of Ti(C,N) and Ni could be observed obviously in all patterns. The peaks of WC and Mo were not found in the XRD pattern. WC and Mo would form a solid solution (Ti, W, Mo)(C,N) with TiC and TiN in the sintering [7], which have a similar crystal structure, lattice constants, and diffraction patterns as Ti(C,N). It is mentionable that as the content of SiC_p_ increased, the diffraction peak of the Ni shifted at a high angle gradually. Calculated values of the lattice constant of the binder phase with a different SiC_p_ addition are provided in Figure 2b. This shows that with an increase of SiC_p_ addition, the lattice constant of the binder phase gradually decreased, while that of the hard phase appeared as an increasing trend. The atomic radii of Ti, Mo, W, and Ni were Ti(1.448 Å) > Mo(1.371 Å) > W(1.363 Å) > Ni(1.246 Å) [8,9], so the dissolution of Ti, Mo, and W into Ni in the sintering would result in the expansion of the binder lattice structure. SiC_p_ inhibited the dissolution of Ti, W, Mo in the binder phase, so the cermets without SiC_p_ exhibited the binder phase with the highest lattice constant and the hard phase with the lowest lattice constant. The composition analysis of the binder by EDAX is shown in Table 2. This shows that the content of Ti, W, and Mo decreased with the increase of the SiC_p_ addition.

The microstructure of cermets with a different SiC_p_ addition is exhibited in Figure 3a–f, in which a typical core-rim structure can be observed [7]. Black core phase was undissolved TiC and TiN. The gray-white inner rim phase was (Ti, W, Mo)(C,N), which was a solid solution rich in W and Mo, formed by the diffusion of W and Mo atoms into Ti(C,N) during the solid-phase sintering stage [10,11]. The gray outer rim phase was a diffused precipitation of the undissolved hard phase into the liquid binder during the liquid-phase sintering stage. The black cores of cermet A were mostly larger than 2 µm in size, and the distribution was extremely uneven, as seen in Figure 3a. When SiC_p_ was added into the cermets, the number of the large-sized cores was obviously depleted. The core-rim structure appeared most complete in cermet C with a 2.0 wt.% SiC_p_ addition, in which the distribution of grain size was also homogeneous. When the SiC_p_ content increased to 2.5 wt.%, the thickness of the gray outer rim phase of the cermet decreased, and the core-rim structure of the cermet became blurred with less integrity. When the SiC_p_ content increased to 3.0 wt.%, the thickness of the gray outer rim phase was further reduced, and the rim partially disappeared. The rim phase played a transition role between the cores and the binders, which optimized the wettability and reduced the interface mismatch between the two phases, then improved the fracture strength of the cermets. However, it is a brittle phase—its thickness should be moderate, otherwise, it will reduce the toughness of cermets [12,13]. Therefore, the completed rim phase with considerable thickness would attribute to the excellent properties, as indicated in Figure 3e. Figure 4a–e shows the grain size distribution of the cermets with different SiC_p_. Obviously, with the increase of SiC_p_, the average grain size decreased from 1.80 µm in cermet A to 1.26 µm in cermet E. Therefore, the addition of SiC_p_ has an obvious effect on the grain refinement of cermet. SiC_p_ would hinder the dissolution of the hard phase components in the binder phase and prevent the dissolved hard phase components from depositing and precipitating on the surface of the undissolved hard phase, restrain the agglomeration and growth of ceramic particles in the sintering process, thus the grain size was decreased and microstructure uniformity of the cermet improved, conducive to improving the strength and toughness of the material.

In order to clarify the role of SiC_p_ in the cermets, EDAX linear analysis was performed and crossed an area with a completed core-rim structure, as seen in Figure 5a, with the results exhibited in Figure 5b. The region could be divided into several parts across the scanning routine, according to different phases and lengths. Core, inner rim, outer rim, and binder phases corresponded from Ⅰ to Ⅳ, respectively. Ti was mainly distributed in the black core, and the binder element Ni decreased sharply in part Ⅲ, which indicated Ni most existed in the binder. The highest content of Mo and W was observed in region Ⅱ, and this was in accordance with the formation of (Ti, W, Mo) (C, N) on the inner rim. Si was gradually decreased with the direction from the binder region to the core region, which meant that Si was mostly reserved for the binder. This would be attributed to the most added SiC_p_ not dissolving into the core region.

The effect of the addition of SiC_p_ on the transverse fracture strength and hardness of Ti(C,N)-based cermets is shown in Figure 6a. As the amount of SiC_p_ increased, the strength was firstly increased and then dropped sharply. The peak value was 2370.3 MPa, corresponding to cermet C with 2.0 wt.% SiC_p_, which was approximately 21.5% higher than the cermet without SiC_p_. This was mainly attributed to the fine grain size, uniform microstructure, and considerable thickness of the rim phase, as shown in Figure 3c. However, an exceeded SiC_p_ addition would lead to an adverse effect on the strength of the cermet, which was mainly due to the incomplete core-rim structure and the large interfacial stress between the binder and cores. On another side, the hardness of the cermets appeared similar to changing tendency with the strength, and the peak value was 88.5 HRA, owned by the one with 2.0 wt.% SiC_p_. The addition of SiC_p_ would result in the elevation of the hardness of the materials due to the increase in the amount of SiC_p_ and a fine grain strengthening effect. However, an exceeded SiC_p_ addition would lead to a decrease in the relative density of the cermets, as seen in Figure 6b, and this was mainly attributed to the poor wettability between the SiC_p_ and binder, thus, the hardness of the cermets was declined.

The fracture morphologies of cermets with different SiC_p_ are illustrated in Figure 7. The dimples formed by the crack propagation in the binder phase and the cleavage planes formed by the crack propagation in the hard phase are clearly observed in Figure 7a–e, which indicates that the fracture was a combination of plastic and brittle fractures. On another side, transgranular and intergranular fractures were both observed in all cermets, which commonly correspond to large grains and small grains, respectively. With the increasing SiC_p_, the number of transgranular fractures reduced, and this further proved that the grain size was refined, which was beneficial in improving the properties of cermets [14]. However, obvious pores were observed in the fracture when the SiC_p_ addition was exceeded. These pores would also affect the transverse fracture strength of the cermet adversely, which is shown in Figure 6a [15].

The influence of SiC_p_ addition on the fracture toughness is shown in Figure 6b. The fracture toughness increased first and then decreased sharply. The highest fracture toughness was achieved according to cermets with 2.0 wt.% SiC_p_, which was 15.2 MPa·m^1/2^. The fracture toughness of cermet is mainly affected by the relative density of the matrix, crack deflection mode. When the content of SiC_p_ was more than 2.0 wt.%, the rapid decrease in fracture toughness was due to the decrease in the relative density of the cermet.

Crack propagation morphologies of cermets with 0.0 wt.%, 2.0 wt.%, and 3.0 wt.% SiC_p_ additions are shown in Figure 8, respectively. For cermet A, the crack extended directly through the large cores, which appeared as a transgranular mode, as seen in Figure 8a. For cermet C with more fine grains, the crack deflection was obviously observed, and the fracture mode appeared to be intergranular. This would render the propagation path tortuous, which could consume more fracture energy in the process and thus promote fracture toughness [16]. For cermet E with excessive SiC_p_, the fracture was also mainly intergranular, while many pores were observed, and this was in accordance with the results in Figure 7d,e. The pores could behave as the source of crack and as the convenient path in the propagation process; then played a reverse role in the strength and toughness of the cermets.

### 3.2. Effect of Sic_p_/Sic_w_ on Cermets

From the above research, it was found that when the SiC_p_ addition amount was 2.0 wt.%, the overall performance was optimal, thus the cermets F–I maintain the total amount of SiC added to 2.0 wt.%.

Morphologies of a cross-section of cermets F–I are shown in Figure 9a–d. A typical core-rim structure was obtained [2]. As seen in Figure 9a, more black cores of large particles existed in cermet F with only 2.0 wt.% SiC_w_. With the ratio of SiC_p_ increased, the microstructure of cermet was refined, and the distribution of microstructure was more uniform, and thus, was similar to the phenomenon shown in Figure 3. This indicated the SiC_p_ was more effective on microstructure refinement than SiC_w_. EDAX point analyses, in Figure 9d, were carried out, and the results are shown in Table 3. The composition of the black core, outer rim, and inner rim was similar to other commonly reported Ti(C,N). The most mentionable point was that Si was mostly presented in a binder and is consistent with the results in Figure 5.

The effects of different ratios of SiC_p_ to SiC_w_ on the mechanical properties of Ti(C,N)-based cermets are shown in Figure 10. This exhibited that the transverse fracture strength increased with the inclined ratio of SiC_w_. The cermets I with 1.5 wt.% SiC_p_ and 0.5 wt.% SiC_w_ had a peak strength of 2520.8 MPa—it was higher than that of cermet C with 2.0 wt.% SiC_p_. The hardness was little influenced by the change of SiC_p_ proportion. The highest fracture toughness of cermets was obtained corresponding to cermet G with 0.5 wt.% SiC_p_ and 1.5 wt.% SiC_w_, and the value was 17.85 MPa·m^1/2^. However, toughness was declined with a further reduction ratio of SiC_w_, and the value was 16.56 MPa·m^1/2^. Even so, it was higher than that of cermet C, which possessed the highest toughness when only SiC_p_ was added, as seen in Figure 6.

Figure 11a–d showed the grain size distribution of the cermets with different ratios of SiC_p_ to SiC_w_. Obviously, with the increase of SiC_p_ and decrease of SiC_w_, the average grain size decreased from 1.68 µm in cermet F to 1.42 µm in cermet I. According to the Hall–Petch formula and dislocation theory [17,18], the smaller the grain size, the more grain boundaries, the greater the dislocation resistance. At the same time, the grain boundary could also prevent the propagation of cracks and improve the mechanical properties of cermets.

After adding SiC_p_, SiC_p_ would better suppress the hard phase particles aggregation to grow up and hinder the dissolution of hard phase components in the binder phase than SiC_w_, and prevent the dissolved hard phase components from deposition and precipitation on the surface of the remained hard phase, thus, as to refine the microstructure of the material. In addition, the cermets I with 1.5 wt.% SiC_p_ and 0.5 wt.% SiC_w_ had a complete core-rim structure, which was conducive to the transverse rupture strength, as seen in Figure 9d.

From Figure 10b, it can be seen that with the SiC_p_ addition, the relative density of the cermets increased. This was due to the cermets with pure SiC_w_—which were easy to agglomerate and not easy to disperse—worsening the density of the cermet. The highest fracture toughness for cermets G with the 0.5 wt.% SiC_p_ and 1.5 wt.% SiC_w_ addition was attributed to the increased relative density. The fracture toughness of cermets decreased with the reduction of SiC_w_, which was due to SiC_w_ combining high strength and high elastic modulus, offering an effective toughening increase for the ceramic matrix composite. The reduction of SiC_w_ reduced the whisker bridging and whisker pulling toughening effect [4,19].

In several pieces of literature [20,21], it had been found that the fracture toughness provided by coarse grains was higher than that of fine grains in the microstructure of cermet due to the binding force provided by the transgranular fracture of coarse grains being greater than that provided by the intergranular fracture of fine grains. The fracture toughness of cermets F, G, and H was greater than that of cermets I, to some extent, because a certain amount of coarse grain was beneficial to the fracture toughness. On the other hand, as shown in Figure 12, the EDS scanning results show the high-multiple fracture morphology of cermet F. In Figure 12a, it can be seen that there were some holes (green arrows) on the fracture, while there was a white projection (red box) around one of the holes. Energy spectrum analysis was conducted, and the results are shown in Figure 12b. According to the EDS results, the white projection contained a lot of Si, and there were some white rods around the hole (red arrows). Thus, these white bars could be SiC_w_, and these holes could be the remainder of the SiC_w_ being pulled out. Thus, the existence of the SiC_w_ improved the fracture toughness of cermets.

Based on the above discussion, suitable amounts of SiC_p_ could significantly refine the grain size of cermets, obtain a complete core-rim structure, and obviously improve the transverse rupture strength of cermets. Nonetheless, the addition of SiC_w_ effectively increased the fracture toughness of cermets. Compared with separate additions of SiC_p_ or SiC_w_, the addition of SiC_p_ and SiC_w_ in appropriate proportions could achieve the purpose of both toughening and strengthening, and thus, grant the best comprehensive mechanical properties.

## 4. Conclusions

In this work, the microstructure, and mechanical properties of Ti(C,N)-based cermets with the addition of the SiC particles (SiC_p_) and SiC whiskers (SiC_w_) were systematically studied, and the results are exhibited below:When SiC_p_ was added into the Ti(C,N)-based cermets independently, the peak value of the mechanical properties was achieved when the addition amount was 2.0 wt.%. The transverse rupture strength, hardness, and fracture toughness of the cermets reached 2370.3 MPa, 88.5 HRA, and 15.2 MPa·m^1/2^, respectively.When SiC_p_ and SiC_w_ were added synergistically, as the ratio of SiC_p_ increased, the grain size of the cermet was decreased, and the density, strength, and hardness were increased, while the fracture toughness was slightly reduced.The cermets with 1.5 wt.% SiC_p_ and 0.5 wt.% SiC_w_ addition appeared to be the best comprehensive properties, and the strength, hardness, and fracture toughness reached 2520.8 MPa, 88.0 HRA, and 16.56 MPa·m^1/2^, respectively. The synergistic strengthening and toughening effect afforded by the reasonable SiC_p_ and SiC_w_ could be confirmed, and the smallest grain size, most uniform, and completed core-rim structure of the cermets were achieved.

## Figures and Tables

**Figure 1 materials-15-02080-f001:**
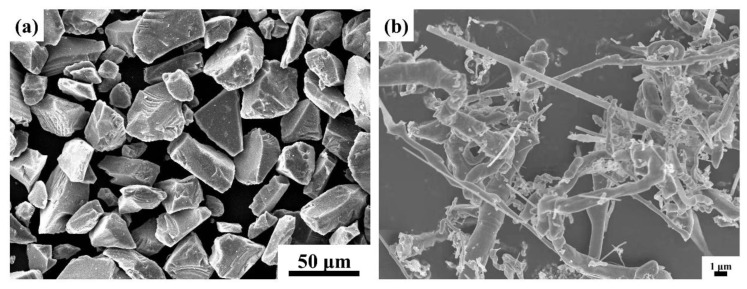
Microscopic morphology of additives: (**a**) SiC_p_; (**b**) SiC_w_.

**Figure 2 materials-15-02080-f002:**
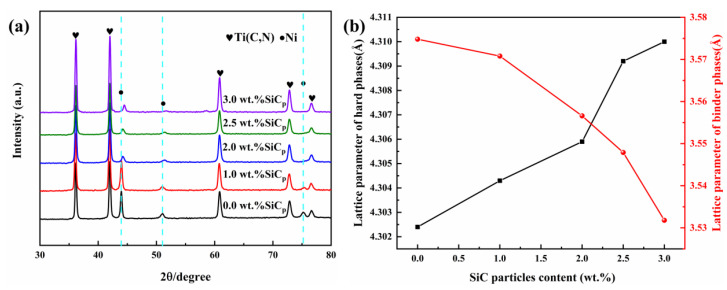
(**a**) Typical XRD patterns of the as-prepared Ti(C,N)-based cermets with various amounts of SiC_p_. (**b**) Effect of SiC_p_ content on the lattice parameter of hard phase and binder phase.

**Figure 3 materials-15-02080-f003:**
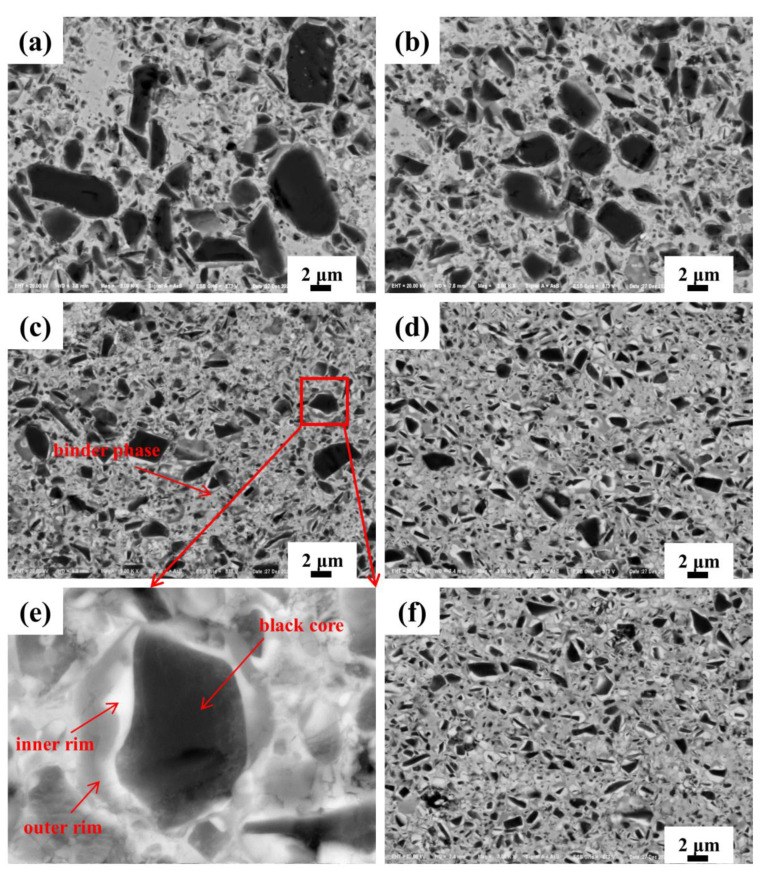
SEM/BSE micrographs of Ti(C,N)-based cermets with different SiC_p_ content: (**a**) 0.0 wt.%; (**b**) 1.0 wt.%; (**c**) 2.0 wt.%; (**d**) 2.5 wt.%; (**e**) a partial enlargement of (**c**). (**f**) 3.0 wt.%.

**Figure 4 materials-15-02080-f004:**
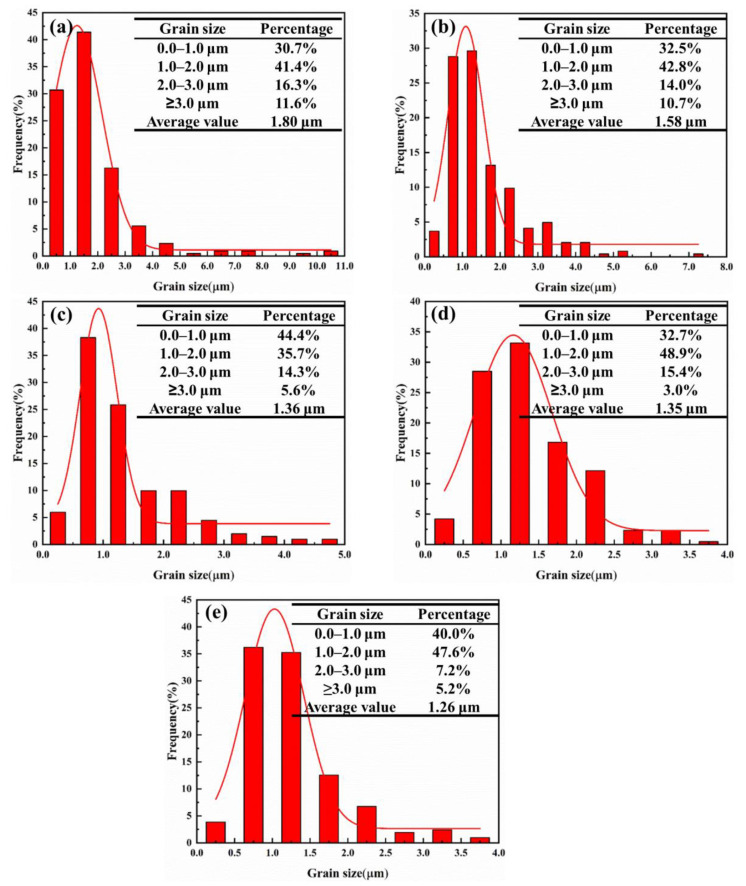
Grain size distribution of the cermets with different SiC_p_ content: (**a**) 0.0 wt.%; (**b**) 1.0 wt.%; (**c**) 2.0 wt.%; (**d**) 2.5 wt.%; (**e**) 3.0 wt.%.

**Figure 5 materials-15-02080-f005:**
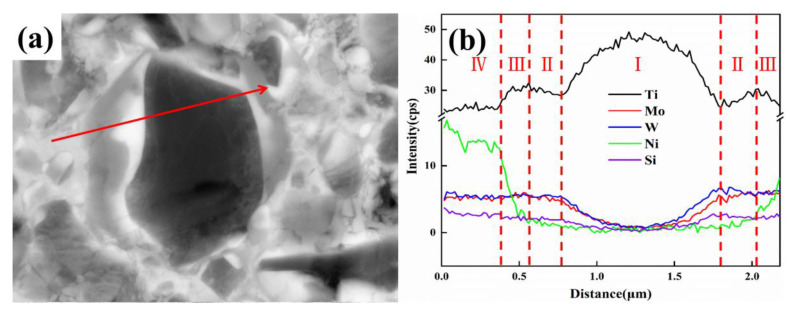
(**a**) The SEM of cermets C with 2.0 wt.% SiC_p_; (**b**) EDS line scanning results.

**Figure 6 materials-15-02080-f006:**
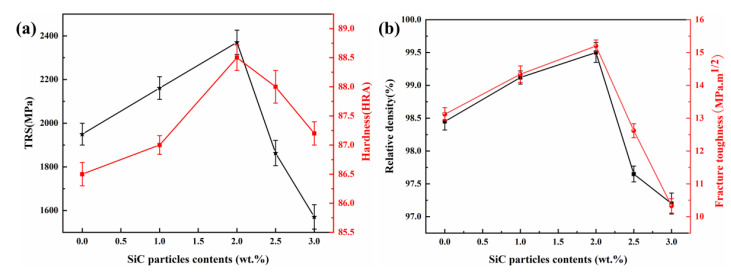
The variation of mechanical properties under different cermets: (**a**) TRS and hardness; (**b**) relative density and fracture toughness.

**Figure 7 materials-15-02080-f007:**
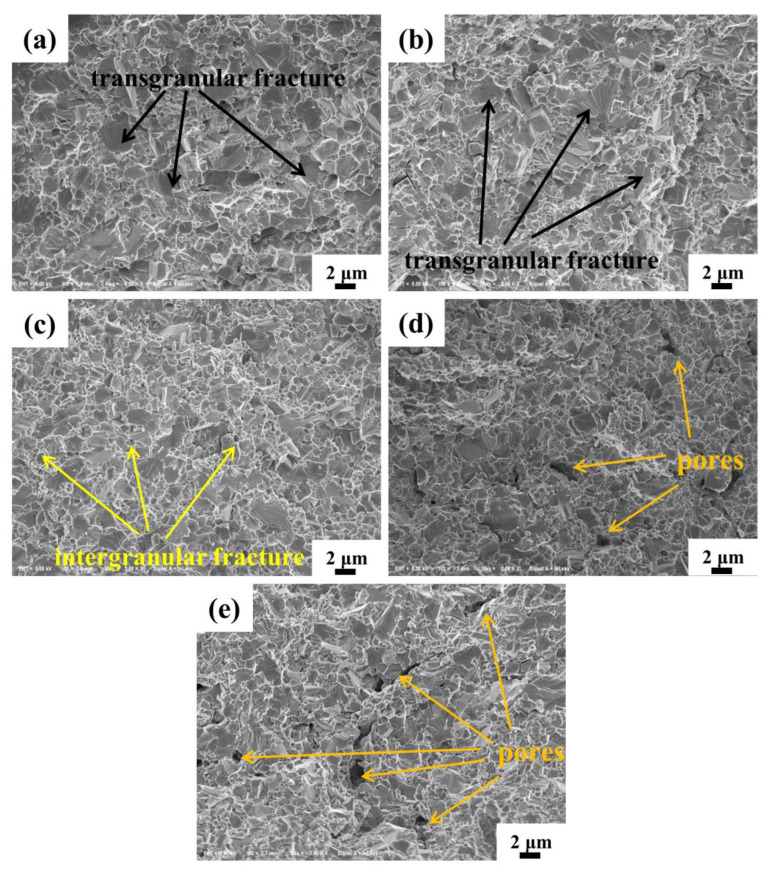
Fracture surface of Ti(C,N)-based cermets with different SiC_p_ content: (**a**) 0.0 wt.%; (**b**) 1.0 wt.%; (**c**) 2.0 wt.%; (**d**) 2.5 wt.%; (**e**) 3.0 wt.%.

**Figure 8 materials-15-02080-f008:**
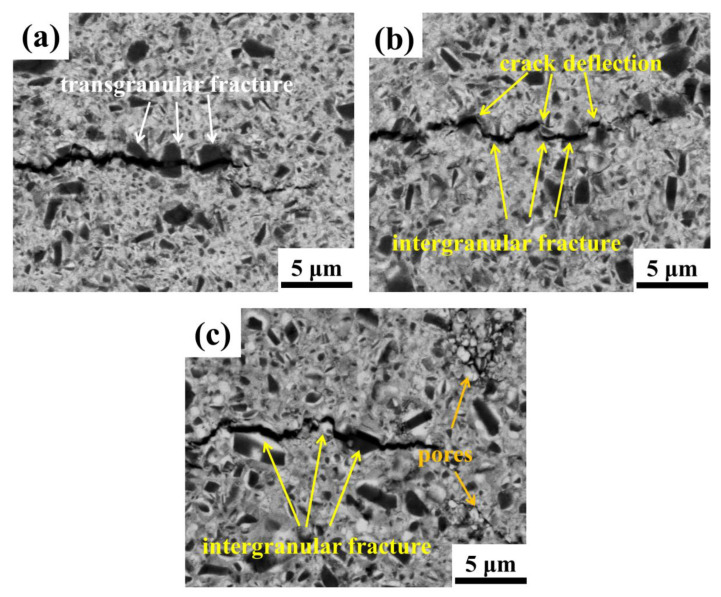
The crack propagation path of Ti(C,N)-based cermets with different SiC_p_ content: (**a**) 0.0 wt.%; (**b**) 2.0 wt.%; (**c**) 3.0 wt.%.

**Figure 9 materials-15-02080-f009:**
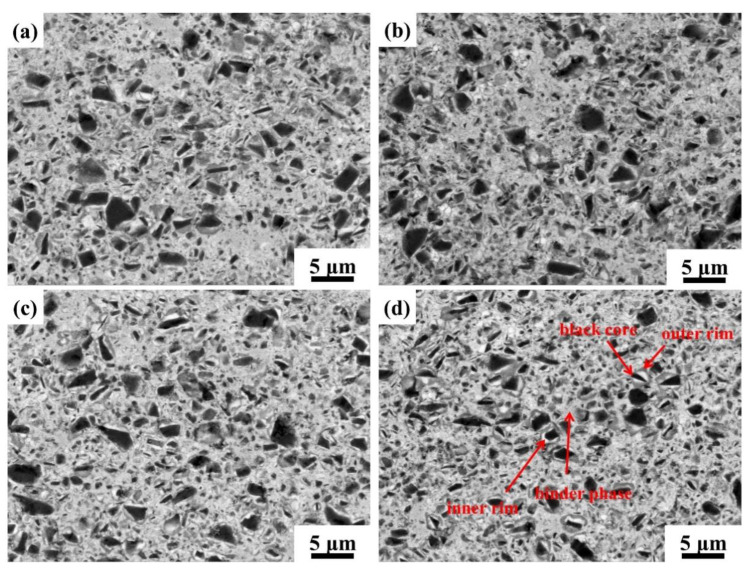
BSE-SEM images of cermets with different ratios of SiC_p_ to SiC_w_: (**a**) 0.0SiC_p_ 2.0SiC_w_; (**b**) 0.5SiC_p_ 1.5SiC_w_; (**c**) 1.0SiC_p_ 1.0SiC_w_; (**d**) 1.5SiC_p_ 0.5SiC_w_.

**Figure 10 materials-15-02080-f010:**
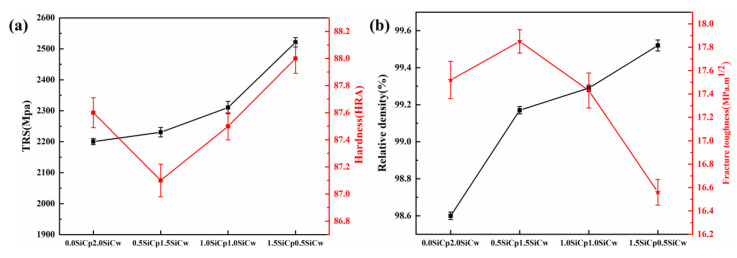
The variation of mechanical properties under different cermets: (**a**) TRS and hardness; (**b**) relative density and fracture toughness.

**Figure 11 materials-15-02080-f011:**
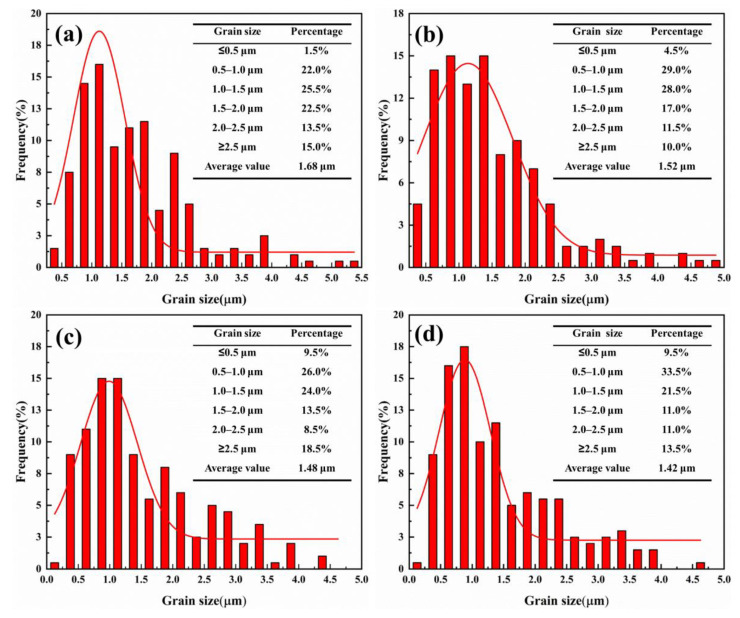
Grain size distribution of the cermets with different ratios of SiC_p_ to SiC_w_: (**a**) 0.0SiC_p_2.0SiC_w_; (**b**) 0.5SiC_p_1.5SiC_w_; (**c**) 1.0SiC_p_1.0SiC_w_; (**d**) 1.5SiC_p_0.5SiC_w_.

**Figure 12 materials-15-02080-f012:**
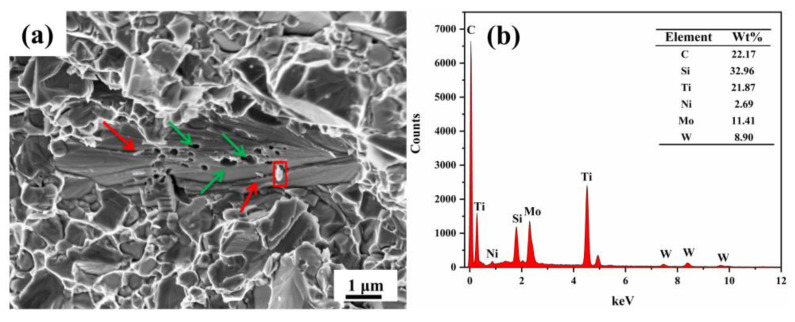
(**a**) The fracture surfaces of cermets F with 2.0 wt.% SiC_w_; (**b**) EDS scanning results.

**Table 1 materials-15-02080-t001:** The composition of M (wt.%).

TiN	Ni	Mo	WC	Cr_3_C_2_	C
10	32	12	10	1	1

**Table 2 materials-15-02080-t002:** Effect of SiC_p_ content on composition of the binder (wt.%).

SiC_p_ Content	w(Ti)	w(W)	w(Mo)
0	10.05	7.89	11.43
1	9.81	6.54	10.21
2	8.02	5.51	8.26
2.5	7.62	5.12	7.86
3	6.23	4.68	6.95

**Table 3 materials-15-02080-t003:** Composition of cermet I with 1.5 wt.% SiC_p_ and 0.5 wt.% SiC_w_ (wt.%).

Elements	Ti	W	Mo	Ni	Si	C
black core	70.94	10.49	11.28	1.26	0	6.03
inner rim	56.87	14.67	18.10	1.31	0	9.05
outer rim	65.15	12.16	13.93	1.26	0	7.50
binder phase	6.42	3.40	3.93	76.69	2.49	7.07

## Data Availability

Data presented in this article are available at request from the corresponding author.

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
