# Peer review of "Synergistic Effect of Sic Particles and Whiskers on the Microstructures and Mechanical Properties of Ti(C,N)-Based Cermets"

_materials, 2022, doi:10.3390/ma15062080_

Round 1
Reviewer 1 Report
Properties of Ti(C,N) Cermets have been extensively studied. Authors propose the synergistic effect of SiC particles and whiskers into their microstructures. This somehow makes the novelty of this work. However, it was hard for me to read since the English is not well employed all along the paper.
Almost everywhere authors speak in past when the verb should be in present. For example "Figure 5 showed"instead of "Figure 5 shows". This must be improved. I suggest to use the help of a native English speaker. Also add the verb in line 235 "results shown in".
Reviewer 2 Report
In this experimental work, the authors present phenomenological observations of improvements in mechanical properties in Ti(C,N)-based cermets induced by addition of SiC nanoparticles and whiskers. The results are interesting. The manuscript is also reasonably well motivated: it addresses the question of whether addition of both SiC nanoparticles and SiC whiskers may enhance both strength and fracture resistance. However, the analyses of results and discussion section are not sufficiently good. The explanations for improvements in mechanical properties are not convincing.
I may give a more definite assessment after revision.
1) The information provided by figures 7 and 8 are not clear. One would like to see a clear role of SiC addition onto fracture resistance. The arrows point at intergranular and transgranular fracture. However, it is very hard to distinguish between the two. For example, the fracture mechanism in Fig.8a looks the same as in Fig.8b and Fig.8c. Looking at the figure, I would not be able to decide whether those are intergranular or transgranular (or both) fracture mechanisms. The authors need to provide more convincing arguments here.
2) Related to point (1), the figures should also clearly indicate the role of SiC. For example, it would be good to indicate SiC "bridges" that hinder crack opening, or similar. For figures 7 and 8, one cannot tell what is the Ti-C-N phase and what is the SiC phase (I doubt that they form a solid solution).
3) Related to comment (1): Fig. 8b claims that there is crack deflection. That crack does not look much different from those of Fig.8a and Fig.8c. How does one see that there is a deflection? This must be clearly indicated / explained.
4) Note that in Fig.8b and 8c, the yellow text reads "interSgranular". The correct term is "intergranular".
5) There are several typos. For example, the header of Sec. 3.1 reads "Effcet". On row 58 on page two, "widly" should be changed to "widely". Please read carefully the entire ms and fix all typos.
6) The hardness is measured in Rockwell scale. It would be highly beneficial if the authors could provide hardness values in GPa, to make them directly comparable to literature values.
7) On page 3, Section 3, the author claims that dissolving ions with (nominally) larger size than Ni leads to an expansion of the binder structure. I am not persuaded of this argument. It is rather "how well" or "how bad" the solute atoms bind to the host matrix which determines whether their will be an expansion or contraction.
8) The Ti(C,N) ceramic phase contains non-negligible amounts of Mo and W. Note that alloying Ti(C,N) with MoC, MoN, WC, or WN does in itself improve plasticity and fracture resistance due to enhanced occupation of metallic electronic states near the Fermi level [see Acta Materialia 59 (2011) 2121–2134 : Supertoughening in B1 transition metal nitride alloys
by increased valence electron concentration] and [Acta Materialia 144 (2018) 376-385: Elastic properties and plastic deformation of TiC- and VC-based pseudobinary alloys]. Thus, the effects of toughening observed by the authors are partly due to enhanced metallic character of the carbonitride solid solution.
Reviewer 3 Report
It is a nice paper on the study of the effect of SiC particles and whiskers on the microstructure and mechanical properties of Ti(C,N) cermets. There are sufficient details given to replicate the proposed experimental procedures and analysis. Discussion of the results is quite clear and scientifically described. I have only few comments before its publication.
Abstract:
Line 16 - leave space between "when 1.5wt%"
Introduction:
- Line 27: what do you mean with "red hardness?"
- Line 55: In order to avoid repetitions I suggest to change the sentences "In view of the characteristics of particle and whisker toughening, the particles and whiskers are used to..." in "In view of the characteristics of particle and whisker toughening, both are used to..."
2.1. Sample preparation:
- Which type of milling media do you use in the ball milling process? describe it
- Describe all the parameter that you use during the sintering process in order to give a possibility to replicate the exactly experiments (vacuum, inert athmosfere, pressureless sintering, heating rate, cooling rate etc..)
3.1. Effect of SiCp on cermets
- Line 113: you write "SiCp exhibited the binder hard phase...." maybe you meant "SiCp exhibited the binder phase...."
- To better follow the results described in Line 126-139, it would be useful to insert in figure 3 the details of the core-rim structures.
- The graphs in figure 4 should have the same scale in order to see better the differences between the materials
- Figure 5a should have a larger magnification relative to the core-rim structure to better observe the trend of Figure 5b. This appoint must be done
- In the grain size distribution analisys, which part of the grain have you mesurement? core? core + rim? only the rim? Some images with high magnification could help to see these aspects
- Insert standard deviations for hardness and toughness values
3.2. Effect of SiCp/SiCw on cermets
- Change "Table 4" in "Table 3"
- Also in this case, for EDX analysis should be better see the structure of rim-core with high magnification if are different respect to samples A-E.
- The starting SiCp and SiCw are 45 microns; how do they shrink so much in size?
- In the samples with wiskers is it possible that elongated wiskers-like structures are not seen? Perhaps a higher magnification SEM image or a SEM image of the sample in the direction perpendicular to the hot pressing could highlight the wiskers' structures
Round 2
Reviewer 2 Report
The authors have satisfactorily addressed my comments.